# Perturbations are not Enough: Generating Adversarial Examples with Spatial Distortions

## Abstract

Deep neural network image classifiers are reported to be susceptible to adversarial evasion attacks, which use carefully crafted images created to mislead a classifier. Recently, various kinds of adversarial attack methods have been proposed, most of which focus on adding small perturbations to input images. Despite the success of existing approaches, the way to generate realistic adversarial images with small perturbations remains a challenging problem. In this paper, we aim to address this problem by proposing a novel adversarial method, which generates adversarial examples by imposing not only perturbations but also spatial distortions on input images, including scaling, rotation, shear, and translation. As humans are less susceptible to small spatial distortions, the proposed approach can produce visually more realistic attacks with smaller perturbations, able to deceive classifiers without affecting human predictions. We learn our method by amortized techniques with neural networks and generate adversarial examples efficiently by a forward pass of the networks. Extensive experiments on attacking different types of non-robustified classifiers and robust classifiers with defence show that our method has state-of-the-art performance in comparison with advanced attack parallels.

## 1 Introduction

Recently, Deep Neural Networks (DNNs) have enjoyed great success in many areas such as computer vision and natural language processing. Nevertheless, DNNs are demonstrated to be vulnerable to *adversarial attacks* (Goodfellow et al., 2014b; Nguyen et al., 2015; Kurakin et al., 2016), which are data samples carefully crafted to be misclassified by DNN classifiers. For example, in image classification, an adversarial example may imperceptibly look like a legitimate data sample in a ground-truth class but misleads a DNN classifier to predict it into a maliciously-chosen target class or any class different from the ground truth. The former and latter are referred to as *targeted attack* and *untargeted attack*, respectively. In addition, adversarial attacks can be categorised into *poisoning attacks* (attacks during the training phase of classifiers) *vs evasion attacks* (attacks during the testing/inference phase) and *whitebox attacks vs blackbox attacks*. For whitebox attacks, the attacker has full access to the model architecture and parameters of a classifier, while those kinds of information are invisible to blackbox attackers. In this paper, we are particularly interested in untargeted, evasion, and whitebox attacks. However, many of the attacks discussed in this paper (including the proposed ones) can be easily adapted into targeted or blackbox settings.

Shown in Goodfellow et al. (2014b), image classifiers based on DNNs (e.g., Convolutional Neural Networks (CNNs) (LeCun et al., 1998)) are susceptible to small but carefully-chosen perturbations. Therefore, significant research efforts have been devoted into *perturbation-based adversarial attacks*, such as those in Goodfellow et al. (2014b); Moosavi-Dezfooli et al. (2016); Carlini & Wagner (2017); Xiao et al. (2018a). Usually, a classifier is more easily fooled with larger perturbations, but this may result in adversarial examples that are less similar to original/real images. In practice, it is equally important that adversarial examples should mislead the classifier as well as "look like" real images. Therefore, the general task for perturbation-based methods can be formulated as misleading a classifier with a minimum amount of perturbation.

It is reported that the intermediate feature maps (convolutional layer activations) in a CNN are not actually invariant to spatial transformations of the input data, due to its limited, pre-defined pooling mechanism for dealing with spatial variations (Jaderberg et al., 2015; Cohen & Welling, 2015; Lenc & Vedaldi, 2015). Therefore, one can imagine generating adversarial examples by incorporating *spatial distortions* to original images. For example, Figure 1a shows that a CNN classifier successfully classifies the digits of MNIST (LeCun & Cortes, 1998). However, it is misled by the adversarial examples with properly-chosen spatial distortions. In contrast, humans are usually less influenced by such distortions. Motivated by this demonstration, by combining spatial distortions with perturbations, we may be able to generate more realistic adversarial examples with smaller perturbations, which can challenge classifiers without affecting human predictions.

To further demonstrate this idea, given some samples of original MNIST images shown in Figure 1b, we apply a widely-used perturbation-based attack, Projected Gradient Descent (PGD) (Madry et al., 2018), with the maximum perturbations of $\epsilon = 0.3$ (the standard setting of the attack), to attack the previously-mentioned classifier, the corresponding adversarial examples of which are shown in Figure 1c. It can be observed that those adversarial examples, though they fool the classifier, can be detected by humans, meaning that one can easily distinguish real and adversarial examples. On the other hand, shown in Figure 1d, the approach introduced in this paper, combines spatial distortions and perturbations to achieve similar attack performance as PGD, but with much smaller perturbation ($\epsilon = 0.1$). Our adversarial examples are clearly less distinguishable from real images by humans.

Motivated by the above idea, in this paper, we propose a new kind of adversarial attack on DNN-based image classifiers, which generalises the conventional perturbation-based attacks with additional spatial distortions. We name our framework **SdpAdv** (**S**patial **d**istortion + **p**erturbation **A**dversary). Specifically, to attack a pretrained classifier, our approach leverages a trainable joint process of two major steps, where it first generates a spatially distorted intermediate image by performing affine-transformations on a real image and then generates the final adversarial image by adding perturbations to the intermediate image.

The proposed SdpAdv has the following appealing properties:

- With the help of spatial distortions, our method is able to achieve state-of-the-art adversarial attack performance with much less perturbations than existing approaches purely based on perturbations, which generates less distinguishable adversarial examples.

- With a differentiable framework, the learning of SdpAdv can be done efficiently in an amortized way, where two neural networks are learned to generate the specific parameters of the spatial distortions and perturbations for an input image. After SdpAdv is trained, it only takes one-step forward propagation in the two neural networks to generate adversarial examples, which enjoys better efficiency in the testing phase.

- As most of existing robust classifier with defences, like those in Samangouei et al. (2018); Matyasko & Chau (2018), are designed to defend against perturbation-based attacks, they can be less effective to our attacks based on spatial distortions (Xiao et al., 2018b).

- Unlike many other whitebox attack methods, which usually require full access to the model structures and parameters of the classifier, SdpAdv only needs to be trained with access to the predicted probabilities of the classifier and directly generates adversarial examples from input images in the testing phase. That is to say, SdpAdv fits the settings of the semi-whitebox attack (Xiao et al., 2018a), which can be more applicable in practice.

To demonstrate the superiority of our proposed method, we conduct extensive comparisons with state-of-the-art adversarial attacks. The experimental results show that SdpAdv is able to achieve better attack performance against both unprotected and robust classifiers. More interestingly, using less perturbation, our adversarial examples are much less distinguishable from real images.

## 2 BACKGROUND AND RELATED WORK

We now introduce the background and related work on adversarial attacks. Suppose that an image $x$ in the ground-truth label $y$ is the input of a neural network classifier $f$. The predicted label of $f$ given $x$ is denoted as $f(x)$. We assume $f(x) = y$ for a well-trained classifier and will use them interchangeably hereafter. The general goal of adversarial attacks is to generate an adversarial example

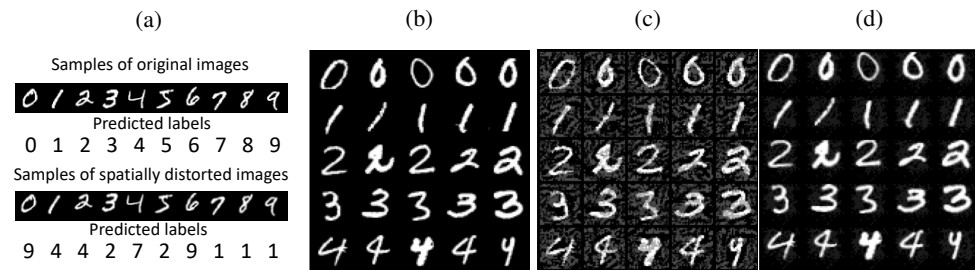

Figure 1: Image classification demo on MNIST dataset. The CNN classifier trained with the training set of the original images achieves 99.15% test accuracy. (a) Original image samples and corresponding spatially distorted images with the predicted labels. (b) Original image samples. (c) The corresponding adversarial examples of PGD with the maximum perturbations of $\epsilon = 0.3$. (d) The corresponding adversarial examples of the attack method proposed in this paper with spatial distortions plus the maximum perturbations of $\epsilon = 0.1$. The accuracies under both kinds of attacks are less than 0.05.

$x_A$, which should "look like" $x$ but change the prediction of $f$, i.e., $f(x) \neq f(x_A)$. Conversely, the goal of a defender is to train a robust classifier to defend against adversarial attacks.

## 2.1 PERTURBATION-BASED ADVERSARIAL ATTACKS

As the name implies, perturbation-based attacks generate $x_A$ by adding small perturbations $\eta$ to $x$: $x_A = x + \eta$. In general, $\eta$ can be constructed by either: $\eta := \arg\max_{\eta':\|\eta'\| \leq \epsilon} \ell_f(x_A, y)$ or $\eta := \arg\min_{\eta':f(x) \neq f(x_A)} \|\eta'\|$, where $\ell_f(\cdot)$ denotes the cross-entropy loss of $f$ and $\|\cdot\|$ can be the $L_\infty$ in accordance with Madry et al. (2018); Athalye et al. (2018) or other norms. As finding the closed-form solution for the above problem can be hard, Fast Gradient Sign Method (FGSM) (Goodfellow et al., 2014b) is a one-step attack that applies a first-order approximation: $\eta = \epsilon \cdot \text{sign}(\nabla_x \ell_f(x, y))$. Several extensions and variants to FGSM have been proposed, such as Randomized FGSM (Tramèr et al., 2018), Basic Iterative Method (BIM) (Kurakin et al., 2016), and Projected Gradient Descent (PGD) (Madry et al., 2018). In addition to FGSM, there are numerous perturbation-based attacks that use different approximations to the above problem. For example, DeepFool (Moosavi-Dezfooli et al., 2016) generates adversarial perturbations by taking a step in the direction of the closest decision boundary in an iterative manner and CW (Carlini & Wagner, 2017) is an attack based on the optimisation of a modified loss function with implicit box-constraints.

Despite their success, the optimisation process of the above methods may have to be done for every test image, which could be inefficient in practice. To alleviate this problem, several attack methods have been proposed to directly generate perturbations by feeding real images into a generator: $\eta = g(x)$, which is usually implemented by neural networks. For example, Adversarial Transformation Networks (ATNs) (Baluja & Fischer, 2018) trains $g$ by minimising the combination of the re-ranking loss and an $L_2$ norm loss, so as to constrain $x_A$ to be close to $x$ in terms of $L_2$. Instead of using an $L_2$ norm, the AdvGAN (Xiao et al., 2018a) attack adopts a Generative Adversarial Network (GAN) (Goodfellow et al., 2014a) framework with a discriminator to encourage the perceptual quality of the generated attacks. Moreover, the Rob-GAN (Liu & Hsieh, 2019) attack also uses a GAN framework, where the discriminator is trained to distinguish between the attacks generated by PGD and those by the generator.

## 2.2 NON-PERTURBATION-BASED ADVERSARIAL ATTACKS

Here we consider non-perturbation-based methods as attacks that do not purely rely on manipulating the pixel values of original images. Compared with perturbation-based attacks, to our knowledge, research on non-perturbation-based methods is relatively rare. Recently, by adding perturbations to the latent space learned by Auxiliary Classifier GAN (AC-GAN) (Odena et al., 2017), the attack in Song et al. (2018) generates adversarial examples for a specific label from scratch without taking a real image as input. Alternatively, there are attacks that apply spatial transformations to original images. Spatial Transformation Method (STM)[1] used in the CleverHans library (Papernot et al., 2018) constructs adversarial candidates by rotation and translation then selects the one that chal-

---

[1] https://cleverhans.readthedocs.io/en/latest/source/attacks.html

lenges the classifier most. As the selection process is non-differentiable, STM relies on the SPSA adversary (Uesato et al., 2018), which is a gradient-free optimization method. Given a real image, stAdv (Xiao et al., 2018b) is differentiable method that finds a flow field, each cell of which captures the transformation direction of one pixel of an input image.

## 3  METHODS

### 3.1  PROBLEM DEFINITION

Here we first present the problem definition of adversarial attacks, which generalises the one of perturbation-based methods, discussed in Section 2. Suppose that a real image and its ground-truth label is denoted by $x \in \mathbb{R}^L$ and $y \in \{1, \cdots, K\}$, respectively, where $L$ represents the image dimension and $K$ is the number of unique labels. We consider a pretrained classifier $f$ taking $x$ as input and implemented with multilayer neural networks, where the last layer has $K$ output units, denoted by $f_l(x)$. The output label of $f(x) : \mathbb{R}^L \to \{1, \cdots, K\}$ is obtained by: $f(x) := \arg\max_l \mathrm{softmax}(f_l(x))$. Given image $x$, we would like to find an adversarial example, $x_A \in \mathcal{O}(x)$, that challenges the classifier $f$ most, i.e., $f(x_A) \neq f(x)$, where $\mathcal{O}(x)$ denotes the set of candidate adversarial images. Given the notation, the most challenging adversarial example $x_A$ can be generated by the following optimisation:

$$x_A := \underset{x' \in \mathcal{O}(x)}{\arg\max} \ell_f(x', y). \tag{1}$$

In particular, $\mathcal{O}(x)$ is usually defined along with the way of generating attacks. For example, for perturbation-based methods, $\mathcal{O}(x)$ can be a $L_2$-ball (or other norms) centred at $x$ with the radius of $\epsilon$: $\mathcal{O}(x) = \{x', \|x' - x\| \leq \epsilon\}$. To make the candidates "perceptually close" to $x$, $\epsilon$ is set to a small value. However, this definition clearly does not fit the proposed approach with spatial distortions, as a small spatial distortion usually keeps an image looking similar but can result in a large $L_2$ difference. Before introducing our definition of $\mathcal{O}(x)$, we present the proposed way of generating adversarial examples in SdpAdv. Given an real image $x$, SdpAdv conducts two major steps: **1)** the spatial distortion step, where a spatial transformation $t : \mathbb{R}^L \to \mathbb{R}^L$: is applied to generate a distorted image $x_T := t(x)$; **2)** the perturbation step, where a perturbation $\eta$ is imposed on $x_T$ to generate the final adversarial example: $x_A := x_T + \eta$.

In the spatial distortion step, we consider the *affine transformation*, which is a widely-used geometric transformation for images. Simply parameterised by a matrix of six real numbers, an affine transformation can be a composition of four linear transformations including *scaling, rotation, shear, and translation*. Given the position of a pixel on a 2D image, $(\mu, \nu)$, an affine transformation with a parameter matrix $\theta$ transforms the pixel into a new position $(\mu', \nu')$ as follows:

$$\begin{bmatrix} \mu' \\ \nu' \\ 1 \end{bmatrix} = \underbrace{\begin{bmatrix} a & b & e \\ c & d & f \\ 0 & 0 & 1 \end{bmatrix}}_{\theta} \begin{bmatrix} \mu \\ \nu \\ 1 \end{bmatrix}, \tag{2}$$

where if $a = d = 1$ and $b = e = c = f = 0$, we get the parameter matrix for the identity transformation, denoted by $\theta_I$. In the case of an image transformation, as the new position $(\mu', \nu')$ can be fractional numbers and so not necessarily lie on the integer image grid, we apply bilinear interpolation to the original image before transformation, following (Jaderberg et al., 2015). With this notation, to generate a distorted image $x_T$ from $x$, we apply an affine transformation $t$ with a parameter matrix $\theta_x$ specific to $x$: $x_T := t_{\theta_x}(x)$. Recall that $x_T$ has to be perceptually close to $x$, thus we would like to avoid over-transforming $x_T$. Therefore, we enforce $\theta_x$ to be in the $L_2$-ball of $\theta_I$: $\|\theta_x - \theta_I\| \leq \gamma$. After the spatial distortion step, we add the perturbation of $\eta_x$ into $x_T$ to generate $x_A$ in the perturbation step: $x_A := x_T + \eta_x$, where $\|\eta_x\| \leq \epsilon$. This step is similar to the conventional perturbation-based methods, except that the perturbations are added into the intermediate image $x_T$ instead of the input image $x$.

Finally, the optimisation problem of SdpAdv can be described as: Given an individual image $x$, finding the optimal $\theta_x$ and $\eta_x$ under the constraints of $\|\theta_x - \theta_I\| \leq \gamma$ and $\|\eta_x\| \leq \epsilon$, which can also be formulated as:

$$\mathbb{E}_{x \sim \mathbb{P}_d} \left[ \max_{\eta' : \|\eta'\| \leq \epsilon, \theta' : \|\theta' - \theta_I\| \leq \gamma} \ell_f(t_{\theta'}(x) + \eta', y) \right], \tag{3}$$

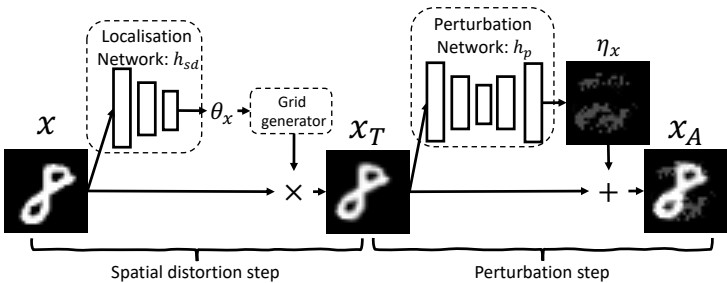

Figure 2: The adversarial generator architecture of SdpAdv.

where $\mathbb{P}_d$ denotes the data distribution of the image training set. It is also noteworthy that the constraints on $\theta$ and $\eta$ define the set of candidate adversarial examples, i.e., $\mathcal{O}(x)$.

## 3.2 AMORTIZED OPTIMISATION SOLUTION

In general, directly solving the problem in Eq. (3) involves the optimisation for every input image $x$, as in many existing perturbation-based algorithms such as CW (Carlini & Wagner, 2017). Such an optimisation process can be challenging and inefficient in practical cases, where real-times attacks may be important. Alternatively, we leverage the amortized optimisation with neural networks to bypass this problem.

First, we introduce two families of neural networks: $\mathcal{H}_{sd} := \{h_{sd} : \theta' := h_{sd}(x) \wedge \|\theta' - \theta_I\| \leq \gamma\}$ and $\mathcal{H}_p := \{h_p : \eta' := h_p(x) \wedge \|\eta'\| \leq \epsilon\}$. In particular, the family of $\mathcal{H}_{sd}$ consists of neural networks with the same network architecture, each of which, $h_{sd}$, takes $x$ as input and outputs the affine parameter matrix $\theta'$ in the $L_2$-ball with $\theta_I$ as the centre. Similarly, $h_p$ outputs the perturbation of $\eta'$. Due to the infinite capacity of neural networks, they can be used to approximate any continuous function up to any level of precision. Therefore, our goal is to find two neural networks $h_{sd}^* \in \mathcal{H}_{sd}$ and $h_p^* \in \mathcal{H}_p$, which are used to approximate the optimisation in Eq. (3), based on the following theorem:

**Theorem 1.** *If the family $\mathcal{H}$ defined above has infinite capacity, the optimization problem in Eq. (3) is equivalent to the following:*

$$\max_{h_{sd} \in \mathcal{H}_{sd}, h_p \in \mathcal{H}_p} \underbrace{\mathbb{E}_{x \sim \mathbb{P}_d} \left[ \ell_f \left( t_{h_{sd}(x)}(x) + h_p(t_{h_{sd}(x)}(x)), y \right) \right]}_{-\mathcal{L}_{adv}}. \tag{4}$$

The proof is in the appendix.

## 3.3 MODEL ARCHITECTURE OF SDPADV

With Theorem 1, we can amortize the optimisation of SdpAdv by training two neural networks: $h_{sd}$ and $h_p$. Equipped with the two neural networks, we are able to build an end-to-end *adversarial generator*, $g$, that takes $x$ as input and outputs $x_A$ by imposing spatial distortions and perturbations: $x_A := g(x)$. The architecture of the proposed generator of SdpAdv is shown in Figure 2. In the spatial distortion step, we adopt the architecture of STN (Jaderberg et al., 2015). Specifically, following Jaderberg et al. (2015), we name $h_{sd}$ to the "localisation network", which takes $x$ as input and outputs the optimal parameter matrix of the affine transformation, $\theta_x$. After that, $\theta_x$ is fed into the "grid generator" to create a sampling grid, which is a set of points where $x$ should be sampled to produce $x_T$ (Jaderberg et al., 2015). In the perturbation step, the adversarial generator takes the output of the previous step, $x_T$, and then adds the optimal perturbation, $\eta_x$ to obtain $x_A$. In particular, $\eta_x$ is generated from $h_p$, named the "perturbation network": $\eta_x := h_p(x_T)$. Finally, the operations of the adversarial generator can be summarised as:

$$g(x) := t_{h_{sd}(x)}(x) + h_p(t_{h_{sd}(x)}(x)). \tag{5}$$

### 3.4 LEARNING OF SDPADV

Given the architecture of the adversarial generator, the learning of SdpAdv is equivalent to that of the two neural networks: $h_{sd}$ and $h_p$, by minimising the loss of $\mathcal{L}_{\text{adv}}$ in Eq. (4). To train the model with stochastic gradient descent (SGD), it is important to show that the model construction is differentiable so that the loss gradients can be backpropagated. It is not hard to see that $h_p$ is trainable, which follows a similar construction to residual neural networks (He et al., 2016). Shown in Jaderberg et al. (2015), the construction of STN also allows the loss gradients to flow back to the grid generator as well as the localisation network $h_{sd}$.

In addition to $\mathcal{L}_{\text{adv}}$, recall that we need to enforce $\|\theta_x - \theta_I\| \leq \gamma$ and $\|\eta_x\| \leq \epsilon$. Therefore, we introduce two regularisation loss functions, respectively, as follows:

$$\mathcal{L}_\theta := \mathbb{E}_{x \sim \mathbb{P}_d}[\|\theta_x - \theta_I\|], \tag{6}$$
$$\mathcal{L}_\eta := \mathbb{E}_{x \sim \mathbb{P}_d}[\max(0, \|\eta_x\| - \epsilon)], \tag{7}$$

where $\|\cdot\|$ is the $L_2$ norm and Eq. (7) is the hinge loss used in Xiao et al. (2018a). In our implementation, after generating $\theta_x$ and $\eta_x$, we apply $\theta_x := \min(\theta_I + \gamma, \theta_x)$ and $\eta_x := \min(\epsilon, \eta_x)$ to keep $x_A$ in the candidate set $\mathcal{O}(x)$ defined by our method.

Inspired by the GAN construction in Xiao et al. (2018a), we also introduce a neural network-based discriminator $d$ to encourage that adversarial images are perceptually close to real images. However, different from Xiao et al. (2018a), our $d$ distinguishes between $x_T$ (positive sample) and $x_A$ (negative sample) and does not care about the spatial distortion step. This is because $x_T$ is expected to naturally "look similar" to the original image $x$ under small affine transformations. Therefore, it is unnecessary to use the discriminator to check this. Following Goodfellow et al. (2014a), the GAN loss is as follows:

$$\mathcal{L}_{\text{GAN}} := \mathbb{E}_{\substack{x \sim \mathbb{P}_d, \\ x_T := t_{h_{sd}(x)}(x)}}[\log D(x_T)] + \mathbb{E}_{\substack{x \sim \mathbb{P}_d, \\ x_T := t_{h_{sd}(x)}(x), \\ x_A := x_A + h_p(x_T)}}[\log(1 - D(x_A))]. \tag{8}$$

In conclusion, the learning of the adversarial generator can be done by minimising the following loss function:

$$\mathcal{L}_g := \mathcal{L}_{\text{adv}} + \alpha\mathcal{L}_\theta + \beta\mathcal{L}_\eta + \lambda\mathcal{L}_{\text{GAN}}, \tag{9}$$

where $\alpha, \beta, \gamma$ are the weight parameters for the losses. In addition, the discriminator can be trained by maximising the GAN loss, similar to Goodfellow et al. (2014a); Xiao et al. (2018a).

### 3.5 COMPARISON TO OTHER METHODS

Among the many related adversarial attack methods discussed in Section 2, we consider Adv-GAN (Xiao et al., 2018a), STM implemented in CleverHans (Papernot et al., 2018), and stAdv (Xiao et al., 2018b) as the most related ones. To our knowledge, SdpAdv is the first technique that combines both spatial distortion and perturbation to generate adversarial examples, while others only consider either of the two attacks. Our method can be viewed as a generalisation to AdvGAN. That is to say, if we set $\gamma$ to 0, then the affine transformations in our method will approach to the identity transformation and our SdpAdv reduces to AdvGAN. Despite the fact that STM and stAdv only consider spatial transformations, the way of conducting spatial transformations in our model is different from theirs. Specifically, STM only allows two kinds of transformations: rotation and translation while affine transformations used in our model are more flexible. More importantly, STM proposes spatially distorted candidates and uses a non-differentiable approach to select adversarial examples from those candidates, while ours is a differentiable method, which is more flexible and easier to train. In stAdv, each pixel in the input image has its specific flow vector to capture the transformation direction of the pixel, making the optimisation of the flow vectors potentially inefficient in practice. In contrast, ours uses one affine transformation that is specific to one image, for all the pixels of that image. Consequently, we only need to optimise for the six cells in the affine parameter matrix for each image, which can be efficiently done by a forward pass in the localisation network.

# 4 EXPERIMENTS

In this section, we conduct experiments to compare the performance of SdpAdv with other state-of-the-art adversarial attack methods on attacking both non-robustified image classifiers (raw classifiers without defences) and robust classifiers with defences.

## 4.1 EXPERIMENTAL SETTINGS

Here we mainly consider the MNIST (LeCun & Cortes, 1998) and Fashion MNIST (Xiao et al., 2017) datasets, each of which consists of 60,000 images of ten classes.

**Settings of the classifiers:** For non-robustified classifiers, we consider: Model-A, a CNN-based classifier with "X-Conv(64, 8×8, 2)-ReLU-Conv(128, 6×6, 2)-ReLU-Conv(128, 5×5, 1)-ReLU-FC(10)-Softmax" and Model-B, a fully-connected (FC) classifier with "X-FC(200)-ReLU-FC(200)-ReLU-FC(10)-Softmax" (Samangouei et al., 2018), where $X$ denotes input layer of image. The two classifiers were pretrained on the standard training set (50,000 images) of MNIST and Fashion MNIST. Additionally, we consider robust classifiers based on the following three state-of-the-art defences: Defense-GAN (Samangouei et al., 2018), Adversarial-Critic (Adv-Critic) (Matyasko & Chau, 2018), and Adversarial-Training (Adv-Train) with FGSM ($\epsilon = 0.3$) (Tramèr et al., 2018). All three robust classifiers share the same model architectures with the non-robustified classifiers but defend adversarial attacks in different ways. We used the original implementations of Defense-GAN[2] and Adv-Critic[3], and implemented Adv-Train ourselves on top of CleverHans.

**Settings of SdpAdv:** In the experiments, we used the following architectures for the adversarial generator ($g$) and discriminator ($d$): the localisation network ($h_{sd}$) with "X-FC(20)-FC(6)"; the perturbation network ($h_p$) with "X-FC(128)-FC(128)-X"; $d$ with "X-FC(64)-FC(32)-FC(2)-Softmax". It is noteworthy that other architectures for the above neural networks can also be used in SdpAdv. In the training phase of SdpAdv, we held out 10,000 images in the training set as our validation set and trained our method on the remaining images in the training set with 100 iterations. We reported the attack results with the best model in the validation set. We initialised $\alpha$ to 5.0 and used a decay factor of 0.8 for every ten iterations until it reached to 0.8. $\beta$ and $\lambda$ were set to 1.0. We set $\gamma$ to 0.3 and varied $\epsilon$ in the range of $\{0.0, 0.1, 0.2, 0.3\}$. If $\epsilon = 0.0$, it means that perturbation is turned off in SdpAdv and we name this variant of our model to "SdAdv". The learning of the adversarial generator and discriminator was done by Adam (Kingma & Ba, 2014) with the learning rate of 0.0005 and 0.00005, respectively.

**Settings of the other attacks:** Here we mainly focus on whitebox attacks, where the attackers have full access to the classifiers. However, our SdpAdv only needs the output logits of the classifier in its training phase and does not require any additional information in the testing phase, similar to AdvGAN (Xiao et al., 2018a). For comparison, we consider the following state-of-the-art perturbation-based attack methods: FGSM (Goodfellow et al., 2014b), PGD (Madry et al., 2018), Momentum Iterative Method (Dong et al., 2018), and AdvGAN (Xiao et al., 2018a). We used the CleverHans (Papernot et al., 2018) implementations of the first three attacks with the standard settings except varying $\epsilon$ in the range of $\{0.1, 0.2, 0.3\}$. We implemented AdvGAN by turning off the spatial distortion step in our SdpAdv. For non-perturbation-based attacks, we consider STM implemented in CleverHans and stAdv (Xiao et al., 2018b) implemented by Dumont et al. (2018)[4]. All the classifiers and attacks (including the proposed ones) were implemented in TensorFlow[5].

## 4.2 RESULTS

For quantitative results, we report the classification accuracies on the test set of the classifiers under the attacks. For attackers, lower accuracy indicates better attack performance. The accuracy results for MNIST and Fashion MNIST are shown in Table 1 and Table 2, respectively.

We have the following remarks on our results: **1)** In general, our proposed SdpAdv performs the best on attacking both non-robustified and robust classifiers in the comparison with other attacks. This

---

[2]`https://github.com/kabkabm/defensegan`

[3]`https://github.com/aam-at/adversary_critic`

[4]`https://github.com/rakutentech/stAdv`

[5]`https://www.tensorflow.org`

Table 1: Classification accuracies on MNIST. The first row shows the accuracies of the classifiers under no attack. The first three attacks (including the proposed SdAdv) are spatial-distortion-based methods. The following four attacks are perturbation-based methods. SdpAdv with $\epsilon > 0.0$ is the proposed method with both spatial distortions and perturbations. For the non-robustified classifiers, $\epsilon$ varies in the range of $\{0.1, 0.2, 0.3\}$ while for the robust classifiers, $\epsilon$ is set to 0.3. In the last column of "sum", we show the summation over the accuracies of all the classifiers with $\epsilon = 0.3$. Best results from the attacks with perturbations are in boldface.

(a) Model A

| Attack \ Classifier | Non-robustified | | | Adv-Critic | Defense-GAN | Adv-Train | Sum |
|---|---|---|---|---|---|---|---|
| No attack | 0.9914 | | | 0.9901 | 0.9914 | 0.9916 | 3.9645 |
| STM | 0.4959 | | | 0.4449 | 0.1434 | 0.9481 | 2.0323 |
| stAdv | 0.1305 | | | 0.9933 | 0.2015 | 0.5388 | 1.8641 |
| SdAdv (ours) | 0.0517 | | | 0.2365 | 0.0476 | 0.074 | 0.4098 |
| $\epsilon$ | 0.1 | 0.2 | 0.3 | 0.3 | 0.3 | 0.3 | |
| FGSM | 0.7788 | 0.3457 | 0.1902 | 0.3595 | 0.8104 | 0.9481 | 2.3082 |
| PGD | 0.4239 | **0.0128** | **0.0059** | **0.0462** | 0.8545 | 0.0926 | 0.9992 |
| MIM | 0.4134 | 0.0911 | 0.0868 | 0.1140 | 0.8032 | 0.1584 | 1.1624 |
| AdvGAN | 0.9915 | 0.7679 | 0.1573 | 0.6820 | 0.3010 | 0.9278 | 2.0681 |
| SdpAdv (ours) | **0.0418** | 0.0259 | 0.0204 | 0.2536 | **0.0266** | **0.033** | **0.3336** |

(b) Model B

| Attack \ Classifier | Non-robustified | | | Adv-Critic | Defense-GAN | Adv-Train | Sum |
|---|---|---|---|---|---|---|---|
| No attack | 0.9831 | | | 0.9817 | 0.9831 | 0.9757 | 3.9236 |
| STM | 0.1939 | | | 0.2044 | 0.0792 | 0.1200 | 0.5975 |
| stAdv | 0.1270 | | | 0.9860 | 0.1207 | 0.3436 | 1.5773 |
| SdAdv (ours) | 0.0502 | | | 0.1363 | 0.0666 | 0.0744 | 0.3275 |
| $\epsilon$ | 0.1 | 0.2 | 0.3 | 0.3 | 0.3 | 0.3 | |
| FGSM | 0.3584 | 0.0901 | 0.0457 | 0.3147 | 0.7710 | 0.8753 | 2.0067 |
| PGD | 0.1198 | **0.0205** | **0.0137** | 0.0920 | 0.8509 | **0.0147** | 0.9713 |
| MIM | 0.1146 | 0.0272 | 0.0211 | 0.1087 | 0.7635 | 0.0373 | 0.9306 |
| AdvGAN | 0.6006 | 0.2432 | 0.0504 | 0.7733 | 0.1110 | 0.2868 | 1.2215 |
| SdpAdv (ours) | **0.0281** | 0.0233 | 0.0223 | **0.0821** | **0.0252** | 0.0741 | **0.2037** |

is particularly demonstrated in the "Sum" column, which can be viewed as a measure of the overall performance. **2)** In comparison with attacks using perturbations, with very small perturbation magnitudes (e.g., $\epsilon = 0.1$), other methods struggle to perform, while with the help of spatial distortions, SdpAdv is able to achieve impressive attack results, generating less noisy attacks without sacrificing performance. **3)** Among the attacks with spatial distortions, our variant, SdAdv, achieves the best attack results in most cases. **4)** We observe that robust classifiers such as Defense-GAN, which are usually designed to defend against perturbations, are much less effective against spatial distortions.

To further study our methods, we compare the test accuracies, $L_2$ norm, and running time of the compared attacks in Table 3. It can be found out that our proposed approaches are able to perform attacks as efficient as FGSM, which is a one-step operation, but with much better performance. It is also noteworthy that the adversarial examples generated by our methods can result in large values of $L_2$ norm but they can look very realistic, shown in Figure 3 and Figure 4.

## 5 CONCLUSION

In this paper, we have introduced a new adversarial attack method, named SdpAdv, named SdpAdv, which generates adversarial examples with both spatial distortions and perturbations. Specifically, given an input image, SdpAdv applies affine transformations to conduct spatial distortions and then adds perturbations to the spatially distorted image to generate the final adversarial example. As a differentiable approach, SdpAdv leverages the amortized optimisation with two neural networks to obtain the optimal parameter of affine transformations and the optimal perturbations, respectively. Extensive experiments of attacking different kinds of non-robustified classifiers and robust classifiers have shown that our method achieves the state-of-the-art performance in the comparison with advanced attack parallels. More importantly, in the use of spatial distortions, our proposed approach can produce more realistic adversarial examples with smaller perturbations, which can challenge classifiers well without affecting human predictions.

Table 2: Classification accuracies on Fashion MNIST.

(a) Model A

| Classifier / Attack | Non-robustified | | | Adv-Critic | Defense-GAN | Adv-Train | Sum |
|---|---|---|---|---|---|---|---|
| No attack | 0.9016 | | | 0.9032 | 0.9016 | 0.9057 | 3.6121 |
| STM | 0.1063 | | | 0.2153 | 0.3548 | 0.1296 | 0.8060 |
| stAdv | 0.0705 | | | 0.8868 | 0.0628 | 0.1930 | 1.2131 |
| SdAdv (ours) | 0.1762 | | | 0.3286 | 0.1372 | 0.1502 | 0.7922 |
| $\epsilon$ | 0.1 | 0.2 | 0.3 | 0.3 | 0.3 | 0.3 | ↘ |
| FGSM | 0.2924 | 0.2494 | 0.2435 | 0.0888 | 0.5661 | 0.8838 | 1.7822 |
| PGD | 0.1789 | 0.1270 | 0.0896 | **0.0286** | 0.6265 | 0.0764 | 0.8211 |
| MIM | 0.2375 | 0.2356 | 0.2356 | 0.0529 | 0.5300 | 0.1054 | 0.9239 |
| AdvGAN | 0.4974 | 0.1239 | 0.0568 | 0.1907 | 0.1238 | 0.1854 | 0.5567 |
| SdpAdv (ours) | **0.0330** | **0.0234** | **0.0121** | 0.1213 | **0.0204** | **0.0444** | **0.1982** |

(b) Model B

| Classifier / Attack | Non-robustified | | | Adv-Critic | Defense-GAN | Adv-Train | Sum |
|---|---|---|---|---|---|---|---|
| No attack | 0.8910 | | | 0.8842 | 0.8910 | 0.8869 | 3.5531 |
| STM | 0.1112 | | | 0.1718 | 0.2817 | 0.0967 | 0.6614 |
| stAdv | 0.2196 | | | 0.8286 | 0.1812 | 0.4614 | 1.6908 |
| SdAdv (ours) | 0.1079 | | | 0.2780 | 0.1298 | 0.1420 | 0.6577 |
| $\epsilon$ | 0.1 | 0.2 | 0.3 | 0.3 | 0.3 | 0.3 | ↘ |
| FGSM | 0.2291 | 0.1674 | 0.1566 | 0.1810 | 0.4236 | 0.8558 | 1.6170 |
| PGD | 0.1157 | 0.0719 | 0.0518 | **0.0731** | 0.5547 | 0.0431 | 0.7227 |
| MIM | 0.1629 | 0.1540 | 0.1529 | 0.0813 | 0.4236 | 0.0515 | 0.7093 |
| AdvGAN | 0.2969 | 0.0703 | 0.0286 | 0.2040 | 0.0605 | 0.1015 | 0.3946 |
| SdpAdv (ours) | **0.0203** | **0.0095** | **0.0075** | 0.1028 | **0.0152** | **0.0392** | **0.1647** |

Table 3: Comparison of test accuracy, $L_2$ norm, and running time Here we report the above metrics for the non-robustified classifier with Model A. The test accuracies are copied from Table 1 and Table 2. We sample 100 images from each of the two datasets and calculate $\mathbb{E}_{\mathbb{P}_d}[\|x_A - x\|]$, which is the $L_2$ norm between the input and adversarial images. In addition, we report the running time (seconds) to generate attacks for those 100 sample images. All the attacks ran in the same machine with the same environment. $\epsilon = 0.3$ is used for the attacks with perturbations, unless stated otherwise.

| Dataset | MNIST | | | Fashion MNIST | | |
|---|---|---|---|---|---|---|
| Metric | Acc | $L_2$ | Time | Acc | $L_2$ | Time |
| FGSM | 0.1902 | 33.42 | 0.05 | 0.2435 | 38.11 | 0.05 |
| PGD | 0.0059 | 25.98 | 0.12 | 0.0896 | 30.52 | 0.13 |
| MIM | 0.0868 | 27.17 | 0.10 | 0.2356 | 31.23 | 0.11 |
| AdvGAN | 0.1573 | 8.29 | 0.01 | 0.0568 | 10.91 | 0.02 |
| STM | 0.4959 | 71.08 | 0.15 | 0.1063 | 57.21 | 0.15 |
| stAdv | 0.1305 | 13.35 | 620.8 | 0.0705 | 6.19 | 609.7 |
| SdAdv | 0.0517 | 120.66 | 0.05 | 0.1762 | 94.23 | 0.05 |
| SdpAdv ($\epsilon = 0.1$) | 0.0418 | 111.7 | 0.06 | 0.0330 | 81.70 | 0.06 |
| SdpAdv ($\epsilon = 0.3$) | 0.0204 | 112.92 | 0.06 | 0.0121 | 44.73 | 0.06 |

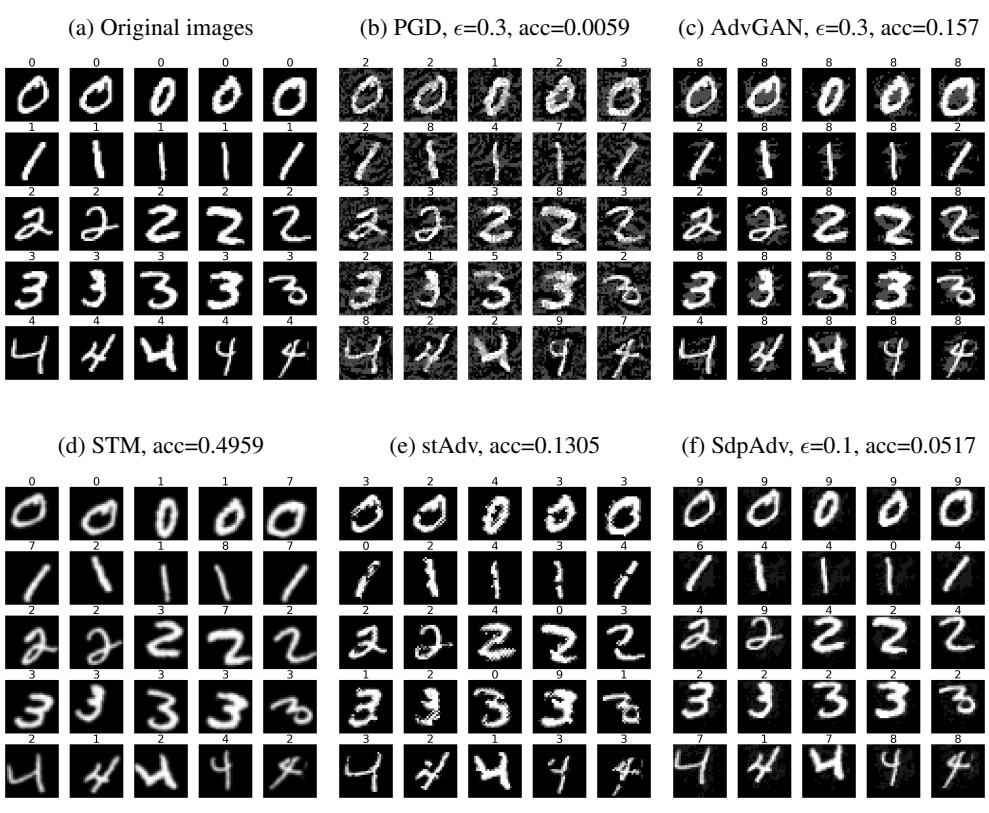

Figure 3: Sample images and their adversarial attacks against Model A in MNIST. The predicted labels of Model A are shown above the images. The test accuracies are copied from Table 1.

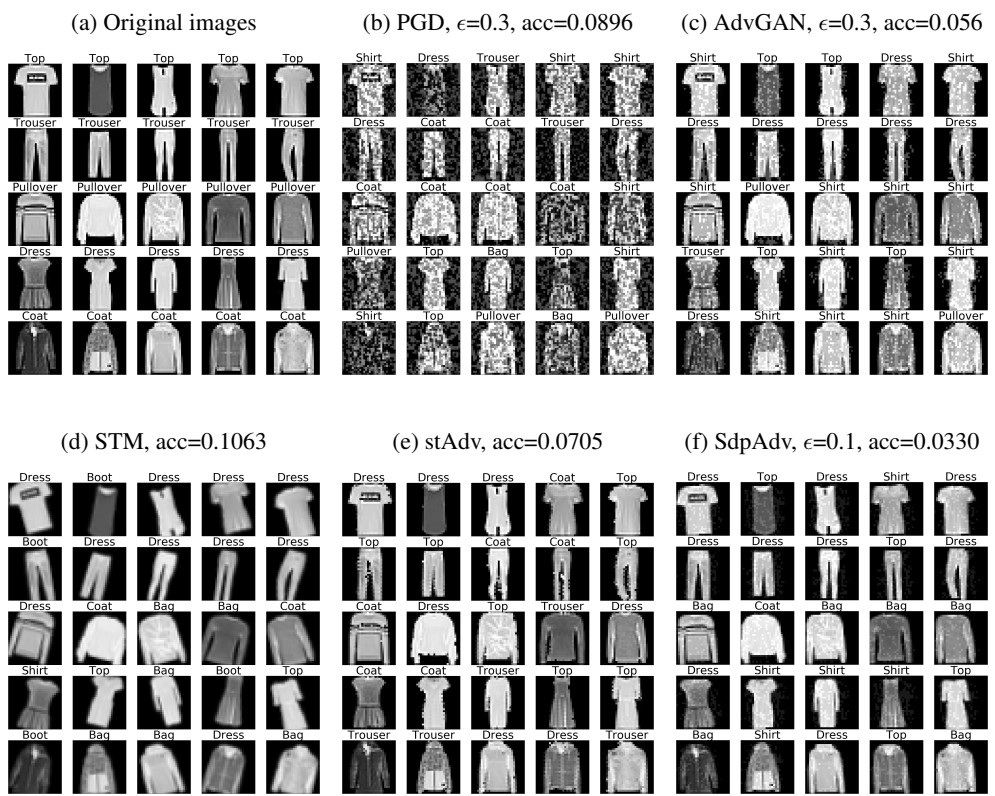

Figure 4: Sample images and their adversarial attacks against Model A in Fashion MNIST. The test accuracies are copied from Table 2.

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

## A  APPENDIX

### A.1  PROOF OF THEOREM 1

*Proof.* Give $x$, suppose $u(x)$ and $v(x)$ are the optimal solution of the affine parameter matrix and perturbation for Eq. (3), respectively. We first prove that $[u(x), v(x)] := h^*(x)$ almost everywhere w.r.t the probability measure $\mathbb{P}_d$, where $h^*(x) := \left[ h^*_{sd}(x), h^*_p \left( t_{h^*_{sd}(x)}(x) \right) \right]$. $h^*(x)$ consists of the optimal solution of Eq. (4). By the definition of $u, v$, it is obvious that

$$\ell_f \left( t_{h^*_{sd}(x)}(x) + h^*_p \left( t_{h^*_{sd}(x)}(x) \right) \right) \leq \ell_f(t_{u(x)}(x) + v(x)).$$

Furthermore, it follows that:

$$\mathbb{E}_{\mathbb{P}_d} \left[ \ell_f \left( t_{h^*_{sd}(x)}(x) + h^*_p \left( t_{h^*_{sd}(x)}(x) \right) \right) \right] \leq \mathbb{E}_{\mathbb{P}_d} \left[ \ell_f(t_{u(x)}(x) + v(x)) \right].$$

Since $\mathcal{H}_{sd}$ has infinite capacity, there exists $h^u_{sd} \in \mathcal{H}_{sd}$ such that $h^u_{sd} = u$. We further define $k(x) = v \left( t^{-1}_{h^u_{sd}(x)}(x) \right)$ and obtain $k \left( t_{h^u_{sd}(x)}(x) \right) = v(x)$. Similarly, with the infinite capacity property of $\mathcal{H}_p$, there exists $h^v_p \in \mathcal{H}_p$ such that $h^v_p = k$

Referring to the definition of $h^*$, we have:

$$\mathbb{E}_{\mathbb{P}_d} \left[ \ell_f \left( t_{h^*_{sd}(x)}(x) + h^*_p \left( t_{h^*_{sd}(x)}(x) \right) \right) \right] \geq \mathbb{E}_{\mathbb{P}_d} \left[ \ell_f \left( t_{h^u_{sp}(x)}(x) + h^v_p \left( t_{h^u_{sp}(x)}(x) \right) \right) \right],$$

$$\mathbb{E}_{\mathbb{P}_d} \left[ \ell_f \left( t_{h^*_{sd}(x)}(x) + h^*_p \left( t_{h^*_{sd}(x)}(x) \right) \right) \right] \geq \mathbb{E}_{\mathbb{P}_d} \left[ \ell_f(t_{u(x)}(x) + v(x)) \right],$$

which further implies the equality:

$$\mathbb{E}_{\mathbb{P}_d} \left[ \ell_f \left( t_{h^*_{sd}(x)}(x) + h^*_p \left( t_{h^*_{sd}(x)}(x) \right) \right) \right] = \mathbb{E}_{\mathbb{P}_d} \left[ \ell_f(t_{u(x)}(x) + v(x)) \right],$$

and also $[u(x), v(x)] := h^*(x)$ almost every w.r.t the probability measure $\mathbb{P}_d$. $\qquad \square$

