# OpenReview forum: "Perturbations are not Enough: Generating Adversarial Examples with Spatial Distortions"
_ICLR.cc/2020/Conference — Reject_

### Official Review · AnonReviewer3 · 2019-10-22
**Official Blind Review #3**

**Rating:** 3

**Review:**

This paper proposes a new adversarial attack method by combining spatial transformations with perturbation-based noises. The proposed method uses two networks to generate the parameters of spatial transformation and the perturbation noise. The whole architecture is trained by a variant of GAN-loss to make the adversarial examples realistic to humans. Experiments on MNIST prove that the proposed attack method can improve the success rate of white-box attacks against several models.

Overall, this paper considers an important problem of adversarial robustness of classifiers, and present a new approach to craft adversarial examples. The writing is clear. However, I have some concerns about this paper.

1. This paper seems to integrate multiple ideas studied before into a single attack method. Perturbation-based adversarial examples, spatial transformation-based adversarial examples, generating adversarial examples based on the GAN loss are all studied before. And the proposed method integrates them together to form a new attack.

2. The experiments are only conducted on MNIST and Fashion MNIST. More experiments on CIFAR-10 and ImageNet can further prove the effectiveness of the proposed method.

3. More robust defense models should be incorporated in experiments, at least the PGD-based adversarial training model (Madry et al., 2018).

**Experience Assessment:**

I have published one or two papers in this area.

**Review Assessment: Checking Correctness Of Derivations And Theory:**

I carefully checked the derivations and theory.

**Review Assessment: Checking Correctness Of Experiments:**

I carefully checked the experiments.

**Review Assessment: Thoroughness In Paper Reading:**

I read the paper at least twice and used my best judgement in assessing the paper.

---

> ### Author Response · Authors · 2019-11-14
> **Many thanks for your precious time and valuable comments.**
>
> Our response is as follows:
>
> 1. Please allow us to re-emphasize the main novelty of our method: We focus on generating adversarial examples that look realistic to humans but also attack the classifier well; We achieve this goal by proposing a generator that conducts both spatial distortions and perturbations. Importantly, the proposed generator is fully differentiable so that we can train it to generate spatial distortions and perturbations jointly. In the joint process, spatial distortions and perturbations are “aware of” each other and “work collaboratively”, so that we are able to use small spatial distortions plus small perturbations to achieve better attack performance.
>
> 2. We are conducting experiments on CIFAR and CelebA. We will try our best to report the results in the rebuttal. If the experiments cannot be concluded by the rebuttal deadline, we will report them in the revised paper.
>
> 3. We have conducted the experiments of adversarial training + PGD, i.e.,  Adv-Train-PGD. The performance results are shown in the following table. It can be observed that Adv-Train-PGD defends well against perturbation-based methods but is less effective than our proposed approaches.
>
>
> +---------------+--------------------------------+--------------------------------+--------------------------------+--------------------------------+
> |               |         MNIST, Model A         |         MNIST, Model B         |     Fashion MNIST, Model A     |     Fashion MNIST, Model B     |
> +---------------+----------------+---------------+----------------+---------------+----------------+---------------+----------------+---------------+
> |               | Adv-Train-FGSM | Adv-Train-PGD | Adv-Train-FGSM | Adv-Train-PGD | Adv-Train-FGSM | Adv-Train-PGD | Adv-Train-FGSM | Adv-Train-PGD |
> +---------------+----------------+---------------+----------------+---------------+----------------+---------------+----------------+---------------+
> |   No attack   |     0.9916     |     0.9915    |     0.9757     |     0.9830    |     0.9057     |     0.9060    |     0.8869     |     0.8854    |
> +---------------+----------------+---------------+----------------+---------------+----------------+---------------+----------------+---------------+
> |      STM      |     0.9481     |     0.4241    |     0.1200     |     0.0413    |     0.1296     |     0.1323    |     0.1112     |     0.1132    |
> +---------------+----------------+---------------+----------------+---------------+----------------+---------------+----------------+---------------+
> |  SdAdv (ours) |      0.074     |     0.0631    |     0.0744     |      0.08     |     0.1502     |     0.1049    |     0.1420     |     0.1850    |
> +---------------+----------------+---------------+----------------+---------------+----------------+---------------+----------------+---------------+
> |      FGSM     |     0.9481     |     0.9710    |     0.8753     |     0.8189    |     0.8838     |     0.7499    |     0.8558     |     0.6076    |
> +---------------+----------------+---------------+----------------+---------------+----------------+---------------+----------------+---------------+
> |      PGD      |     0.0926     |     0.9427    |     0.0147     |     0.7419    |     0.0764     |     0.6619    |     0.0431     |     0.5140    |
> +---------------+----------------+---------------+----------------+---------------+----------------+---------------+----------------+---------------+
> |      MIM      |     0.1584     |     0.9358    |     0.0373     |     0.7201    |     0.1054     |     0.5987    |     0.0515     |     0.4401    |
> +---------------+----------------+---------------+----------------+---------------+----------------+---------------+----------------+---------------+
> |     AdvGAN    |     0.9278     |     0.9906    |     0.2868     |     0.8680    |     0.1854     |     0.3762    |     0.1015     |     0.1387    |
> +---------------+----------------+---------------+----------------+---------------+----------------+---------------+----------------+---------------+
> | SdpAdv (ours) |      0.033     |     0.0752    |     0.0741     |     0.092     |     0.0444     |     0.1113    |     0.0392     |     0.1081    |
> +---------------+----------------+---------------+----------------+---------------+----------------+---------------+----------------+---------------+

---

### Official Review · AnonReviewer2 · 2019-10-23
**Official Blind Review #2**

**Rating:** 1

**Review:**

This paper builds upon the work of AdvGAN and proposes to add spatial transformations on top of it. The resulting attacking framework is demonstrated to outperform AdvGAN on attacking several defense approaches, such as Defense-GAN, AdvCritic and adversarial training. Compared to previous approaches on generating spatially transformed adversarial examples, this approaches amortizes the attacking procedure and can produce spatially transformed adversarial examples much faster. This approach also simultaneously combine spatial transformations and perturbations to make the attack stronger.

I cannot recommend acceptance of this paper because of several reasons:

- The idea is not novel enough. It is simply an A + B paper where A = AdvGAN and B = spatial transformer networks. The idea of adversarial attacks with spatial distortion is not the innovation of this paper and has been proposed and extensively studied by many previous papers. This paper does not have additional innovation and does not lead to additional insight that can warrant an acceptance at ICLR.

- The general narrative of this paper is misleading. The title seems to indicate this paper is the first to discover the importance of considering spatial perturbations, which is misleading. There is no mention of previous work on spatial transformation attacks in either the abstract nor the introduction (except at the very last). The introduction simply analyzes some well-known phenomenon in the literature, does not place this work well in the literature (even true in the related work section as well), and can mislead readers in believing that this work was the first to realize the importance of spatial transformation attacks.

- Theorem 1 is a vacuous statement. It is automatically true based on the universal approximation assumption of neural networks. Including the statement of Theorem 1 is decorative and a waste of space.

- The experiments are not convincing. For example, there is no \gamma value reported in the tables. Since \gamma is as important as \epsilon in the proposed attacking method, the missing of this important variate is suspicious. Also the adversarial training only uses FGSM not PGD. The Defense-GAN is already shown not robust by Athalye et al. and cannot be considered as one of the state-of-the-art defenses.

**Experience Assessment:**

I have published one or two papers in this area.

**Review Assessment: Checking Correctness Of Derivations And Theory:**

I did not assess the derivations or theory.

**Review Assessment: Checking Correctness Of Experiments:**

I assessed the sensibility of the experiments.

**Review Assessment: Thoroughness In Paper Reading:**

I made a quick assessment of this paper.

---

> ### Author Response · Authors · 2019-11-14
> **Many thanks for your precious time and valuable comments.**
>
> Our responses are as follows:
>
> 1. Novelty:
>
> 	a. Please note that we did not claim the idea of “introducing spatial distortions into adversarial attacks” as our innovation. Instead, we have proposed a new approach with spatial distortions, which is better than existing spatial distortion based attacks.
>
> 	b. We are aware of the existing study in spatial distortion based attacks; the most related ones to ours are discussed in Sections 2.2 and 3.5 in our paper; more importantly, we have compared our approach to this in our experiments.
>
> 	c. The novelties of our paper compared with others are discussed in Sections 3.5 and please allow us to re-emphasize the main novelty of our method: We focus on generating adversarial examples that look realistic to humans but also attack the classifier well; We achieve this goal by proposing a generator that conducts both spatial distortions and perturbations; Importantly, the proposed generator is fully differentiable so that we can train it to generate spatial distortions and perturbations jointly; In the joint process, spatial distortions and perturbations are “aware of” each other and “work collaboratively”, so that we are able to use small spatial distortions plus small perturbations to achieve better attack performance.
>
> 	d. We undertook a comprehensive search of the related literature of spatial distortion based attacks. However, we may have missed some papers. It would be great if the reviewer could point out any additional related papers to help us improve our paper. We are more than happy to discuss and compare these papers with our paper.
>
> 2. “Misleading”:
>
> It is unfortunate that the reviewer misunderstood our main claims. We did not make any claims that “we are the first to use spatial distortions in adversarial attacks.” On the contrary, we have compared the recent advances in spatial distortions in detail by means of both discussions (see Sections 2.2 and 3.5) and experiments. As the writing style and presentation order can be very subjective, we respectfully disagree on judging our paper by using them as major points.
>
> 3. Theory:
>
> We will remove the theory as suggested.
>
> 4. Experiments:
>
> 	a. \gamma controls the magnitude of the spatial distortion. We have conducted experiments using varying \gamma values for the proposed methods, with performance results shown in the following table. Please also note that we have released the code to the community so as to reproduce our results. It can be seen that SdAdv is relatively sensitive to \gamma as it only uses spatial distortions. However, SdpAdv is less sensitive to \gamma as perturbations “are aware of” spatial distortions and will help attack.
>
> +--------+--------+--------------------+--------+--------+
> | \gamma |   0.1  | 0.3 (in the paper) |   0.5  |   1.0  |
> +--------+--------+--------------------+--------+--------+
> |             MNIST, Model A, Non-robustified            |
> +--------+--------+--------------------+--------+--------+
> |  SdAdv | 0.7446 |       0.0517       | 0.0352 | 0.0417 |
> +--------+--------+--------------------+--------+--------+
> | SdpAdv |  0.034 |       0.0204       | 0.0131 | 0.0130 |
> +--------+--------+--------------------+--------+--------+
> |             MNIST, Model B, Non-robustified            |
> +--------+--------+--------------------+--------+--------+
> |  SdAdv | 0.3892 |       0.0502       | 0.0501 | 0.0477 |
> +--------+--------+--------------------+--------+--------+
> | SdpAdv | 0.0117 |       0.0233       | 0.0223 | 0.0222 |
> +--------+--------+--------------------+--------+--------+
> |         Fashion MNIST, Model A, Non-robustified        |
> +--------+--------+--------------------+--------+--------+
> |  SdAdv | 0.4213 |       0.1762       | 0.1369 | 0.1285 |
> +--------+--------+--------------------+--------+--------+
> | SdpAdv | 0.0221 |       0.0121       | 0.0225 | 0.0224 |
> +--------+--------+--------------------+--------+--------+
> |         Fashion MNIST, Model B, Non-robustified        |
> +--------+--------+--------------------+--------+--------+
> |  SdAdv | 0.2976 |       0.1079       | 0.1299 | 0.1210 |
> +--------+--------+--------------------+--------+--------+
> | SdpAdv | 0.0158 |       0.0075       | 0.0120 | 0.0125 |
> +--------+--------+--------------------+--------+--------+
>
>
> 	b. We have conducted experiments with adversarial training + PGD i.e., Adv-Train-PGD. Due to the character limits of our response, please find the results in our response to Reviewer 3.

---

### Official Review · AnonReviewer1 · 2019-10-23
**Official Blind Review #1**

**Rating:** 3

**Review:**

The paper introduces a new approach to generate adversarial examples for deep classifiers. As opposed to the majority of work on adversarial attack models, which generally limit the attacker on pixel-space distortions measured with respect to an Lp norm, the authors here consider a slightly more general attack model that is a combination of an affine transformation and additive L2 perturbation of the input example.

Finding optimal attacks for this model can be non-trivial (standard due to the highly nonlinear coupling between the affine parameters and the additive perturbation), so the authors instead propose training a surrogate neural network that generates the attack affine-transformation and distortion- parameters sequentially. This can, in principle, be done in a traditional supervised training setup; however, to force the adversarial images to look perceptually close to natural looking images, the authors throw a discriminator loss on top, and train the attack generator network adversarially.

The paper is well-written in general, the idea is intuitive, and the experiments are well-described. However, I have a few concerns that lead to me to give a low score (at least in the first round of reviews).

- Novelty.
Leveraging spatial distortions (or other visually meaningful transformations) to generate adversarial attacks is not a new idea, but the authors seem to have been unaware of this very large body of work. See, for example:
** Engstrom et al, "Exploring the landscape of spatial robustness", ICML 2019
** Poovendran et al, "Semantic adversarial examples", CVPR 2018
** Ho et al, "Catastrophic Child's Play", CVPR 2019
** Joshi et al, "Semantic adversarial attacks", ICCV 2019
among many others.

Using GAN-like transformation models to generate attacks is also not a new idea. A few of the above papers use this approach, and the authors refer to a few other such papers as well.

So as such, the conceptual novelty of the contribution seems to be low (beyond the specific choice of combining affine and L2 perturbations).

- Experimental evaluation.
The authors do a commendable job thoroughly laying out the experimental setup. However, a couple of red flags emerge in the experiments. First, why not look at L-infty perturbations (as opposed to L2)? Second, why not test on more challenging datasets (CIFAR, CelebA, etc) as opposed to simple black/white datasets such as MNIST/Fashion-MNIST? One would imagine that the smaller, simpler datasets are easier to optimize for, and therefore the "amortized" attack generator networks are not necessary here.

- Weakness of theoretical part.
I am not sure the theorem is saying anything strong or useful (since the underlying transformer neural network is assumed to possess infinite capacity). I would suggest just removing it.


**Experience Assessment:**

I have published one or two papers in this area.

**Review Assessment: Checking Correctness Of Derivations And Theory:**

I carefully checked the derivations and theory.

**Review Assessment: Checking Correctness Of Experiments:**

I assessed the sensibility of the experiments.

**Review Assessment: Thoroughness In Paper Reading:**

I read the paper at least twice and used my best judgement in assessing the paper.

---

> ### Author Response · Authors · 2019-11-14
> **Many thanks for your precious time and valuable comments.**
>
> Our responses are as follows:
>
> 1. Novelty and related work:
>
> 	a. Our proposed attack falls into the category of “spatial distortion based attacks”, which also belongs to a more general research line of “adversarial attacks for images with visually meaningful transformations” (e.g.  “colour-shifting” in “semantic adversarial examples”). Please note that we focus on the category of “spatial distortion based attacks.'' In this category, we believe that our approach has significant novelties and advantages over others in the same category in both effectiveness and efficiency. This is discussed in Sections 2.2 and 3.5 of our paper.
>
> 	b. Please allow us to re-emphasize the main novelty of our method: We focus on generating adversarial examples that look realistic to humans but also attack the classifier well; We achieve this goal by proposing a generator that conducts both spatial distortions and perturbations; Importantly, the proposed generator is fully differentiable so that we can train it to generate spatial distortions and perturbations jointly; In the joint process, spatial distortions and perturbations are “aware of” each other and “work collaboratively”, so that we are able to use small spatial distortions plus small perturbations to achieve better attack performance.
>
> 	c. Thanks for pointing out the interesting papers, which we will add as the references for our paper. For those papers, we have the following discussions:
>
> 		i. “Exploring the landscape of spatial robustness”, ICML19: This paper falls into the same category (spatial distortion based attacks) as ours. Actually, in our discussions and experiments, we compared a spatial distortion method called the “Spatial Transformation Method (STM)”, which is implemented in Cleverhans and is exactly the method proposed in “Exploring the landscape of spatial robustness”. We were unaware of this paper because Cleverhans did not give a reference to this method. We re-summarise the differences between ours and STM: STM only allows translations and rotations while ours allows all kinds of affine transformations; STM uses grid searches to find the optimal translation and rotation while ours is a differentiable method can be jointly trained with perturbation-based methods; STM takes grid searches for every test sample which can be inefficient, while ours only takes a pass of neural networks to conduct attacks (Shown in Table 3 of the paper, where our technique is 3 times faster than STM, even when including perturbations).
>
> 		ii. “Catastrophic Child's Play", CVPR19: This paper only considers random affine transformations with rotations less than 15 degrees while ours optimises the affine transformations according to the loss of the classifier.
>
>  		iii. "Semantic adversarial examples", CVPR18 and "Semantic adversarial attacks", ICCV19: Spatial distortions are not used in these papers, so they might be relatively less related to ours. But we agree that they fall into the general area of "adversarial attacks for images with visually meaningful transformations."
>
> 2. Experimental evaluation:
>
> We are conducting experiments on CIFAR and CelebA. We will try our best to report the results in the rebuttal. If the experiments cannot be concluded by the rebuttal deadline, we will report them in the revised paper.
>
> 3. Weakness of theoretical part:
>
> We will remove the theory as suggested.

---

### Public Comment · ~Dimitris_Tsipras1 · 2019-10-04
**Prior work on combining spatial distortions with pixel-wise perturbations**

I wanted to bring to your attention our work studying spatial distortions---rotations and translations---and their combination with standard pixel-based perturbations: "Exploring the landscape of spatial robustness" (ICML'19, https://arxiv.org/abs/1712.02779). Specifically, we consider an adversary that tries all possible spatial transformations (through exhaustive grid search) and then applies a standard PGD attack on top.

Also, I believe it is worth mentioning "Manitest: are classifiers really invariant?" (BMVC'15, https://arxiv.org/abs/1507.06535) which---to the best of my knowledge---is the first work studying the robustness of deep networks to spatial transformations.

---

> ### Author Response · Authors · 2019-10-06
> **Thanks for the reference**
>
> Dear Dimitris,
>
> Thanks a lot for pointing out this interesting and related paper, which we weren't aware of at the time of working on our submission.
>
> After a quick look at this ICML19 paper, we find that in terms of attacks, there are several differences between ours and the ICML19 one. For example, the ICML19 one conducts rotation and translation to generate an adversarial example by doing optimisation given a test sample, while ours uses an amortized way, which learns a neural network to conduct rotation, translation, scaling, and shear. Moreover, ours is a joint amortized process (with two neural networks), that combines spatial distortions and perturbations, aiming to generate realistic adversarial examples with fewer perturbations. While the ICML19 one seems to use two separate steps for PGD and grid search for rotations and translations, respectively. We will discuss more on differences as well as connections between the two attacks in the updated version of our paper.
>
> We also appreciate that the code of the ICML19 paper is released. Therefore, we are currently doing a comparison with the attack and defence methods introduced in the ICML19 paper and will provide a detailed discussion in the updated version of our paper.
>
> Besides, thanks for pointing out the BMVC paper as well, which will be added to our reference.
>
> Thanks again,
> Paper 1706 Authors

---

### Public Comment · ~Vu_Nguyen1 · 2019-10-09
**Related work in generating distortions under affine transformation**

Great work addressing an important problem in deep learning !

I would like to mention a closely related work [1] that uses Bayesian optimisation (BO) to sequentially suggest an attack to make the deep models failed. The attacks used in [1] has considered different ways of affine transformation including translation, rotation, shearing. Using BO in [1] will be sample-efficient in making the attacks. That is, the targetted deep model will be failed under less number of attacks.

The problem definition is quite similar. That is, the objective function defined in Eq1, used to select the attack $x_A$, in the current paper is similar to objective function defined in Eq1 in [1]. Having said that, the search spaces considered in two papers are different. Particularly, the Eq1 in [1] considered the parameter space of the attacks (such as the shearing parameters) while Eq1 in this paper considered the L2-ball in the raw feature space.

[1] Gopakumar, Shivapratap, et al. "Algorithmic assurance: an active approach to algorithmic testing using Bayesian optimisation." Advances in Neural Information Processing Systems. 2018.

NB: The source code is also available.

---

### Decision · Program_Chairs · 2019-12-19

**Decision:**

Reject

**Comment:**

The method proposed and explored here is to introduce small spatial distortions, with the goal of making them undetectable by humans but affecting the classification of the images. As reviewers point out, very similar methods have been tested before. The methods are also only tested on a few low-resolution datasets.

The reviewers are unanimous in their judgement that the method is not novel enough, and the authors' rebuttals have not convinced the reviewers or me about the opposite.